# Mitochondrial Dysfunction Underlies Cardiomyocyte Remodeling in Experimental and Clinical Atrial Fibrillation

**DOI:** 10.3390/cells8101202

**Published:** 2019-10-05

**Authors:** Marit Wiersma, Denise M.S. van Marion, Rob C.I. Wüst, Riekelt H. Houtkooper, Deli Zhang, Natasja M.S. de Groot, Robert H. Henning, Bianca J.J.M. Brundel

**Affiliations:** 1Department of Physiology, Amsterdam UMC, Vrije Universiteit Amsterdam, Amsterdam Cardiovascular Sciences, 1081 HV Amsterdam, The Netherlandsd.zhang@amsterdamumc.nl (D.Z.); 2Laboratory for Myology, Department of Human Movement Sciences, Faculty of Behavioral and Movement Sciences, Vrije Universiteit Amsterdam, Amsterdam Movement Sciences, 1105 AZ Amsterdam, The Netherlands; r.wust@vu.nl; 3Laboratory Genetic Metabolic Diseases, Amsterdam UMC, University of Amsterdam, Amsterdam Gastroenterology and Metabolism, Amsterdam Cardiovascular Sciences, 1105 AZ Amsterdam, The Netherlands; r.h.houtkooper@amsterdamumc.nl; 4Department of Cardiology, Erasmus Medical Center, 3015 DG Rotterdam, The Netherlands; n.m.s.degroot@erasmusmc.nl; 5Department of Clinical Pharmacy and Pharmacology, University Medical Center Groningen, 9700 RB Groningen, The Netherlands; r.h.henning@umcg.nl

**Keywords:** atrial fibrillation, mitochondria, MCU, Ru360, SS31

## Abstract

Atrial fibrillation (AF), the most common progressive tachyarrhythmia, results in structural remodeling which impairs electrical activation of the atria, rendering them increasingly permissive to the arrhythmia. Previously, we reported on endoplasmic reticulum stress and NAD^+^ depletion in AF, suggesting a role for mitochondrial dysfunction in AF progression. Here, we examined mitochondrial function in experimental model systems for AF (tachypaced HL-1 atrial cardiomyocytes and *Drosophila melanogaster*) and validated findings in clinical AF. Tachypacing of HL-1 cardiomyocytes progressively induces mitochondrial dysfunction, evidenced by impairment of mitochondrial Ca^2+^-handling, upregulation of mitochondrial stress chaperones and a decrease in the mitochondrial membrane potential, respiration and ATP production. Atrial biopsies from AF patients display mitochondrial dysfunction, evidenced by aberrant ATP levels, upregulation of a mitochondrial stress chaperone and fragmentation of the mitochondrial network. The pathophysiological role of mitochondrial dysfunction is substantiated by the attenuation of AF remodeling by preventing an increased mitochondrial Ca^2+^-influx through partial blocking or downregulation of the mitochondrial calcium uniporter, and by SS31, a compound that improves bioenergetics in mitochondria. Together, these results show that conservation of the mitochondrial function protects against tachypacing-induced cardiomyocyte remodeling and identify this organelle as a potential novel therapeutic target.

## 1. Introduction

Atrial fibrillation (AF) is the most common sustained clinical tachyarrhythmia and is associated with increased mortality and morbidity [1,2]. Its incidence is age-related and expected to rise due to the aging population. Consequently, AF will contribute significantly to the socioeconomic burden [3]. Due to its sustained nature, most patients progress from paroxysmal AF to persistent and longstanding persistent AF [2]. Importantly, therapy of (longstanding) persistent AF has high failure rates, with 20–60% of patients showing recurrence of AF within three months after ablation or electrical cardioversion [4,5,6,7]. Therapy failure in AF is related to the presence of structural remodeling of the myocardium, which, in turn, impairs electrical activation of the atria (“electropathology”) [8]. Therefore, recent research is increasingly directed at revealing the pathways underlying AF-induced cardiac structural remodeling in order to develop more mechanism-directed AF therapies.

Recently, we documented that endoplasmic reticulum (ER) stress [9] and a significant reduction in nicotinamide adenine dinucleotide (NAD^+^) levels [10] underlies AF-induced cardiac structural remodeling. The ER is in close contact with mitochondria through the mitochondria-associated membranes and it is known that via this way stress from the ER may be propagated to the mitochondria [11]. NAD^+^ is essential for mitochondrial redox reactions to produce ATP and thereby plays a role in the energy metabolism, particularly in cells with high metabolic activity, such as cardiomyocytes [12]. Therefore, in light of this information, mitochondrial function may be compromised in AF.

Although the role of mitochondria in AF pathogenesis is not clear, it is known that mitochondria play a role in the electropathology of cardiac arrhythmias. Especially mitochondrial Ca^2+^ overload has detrimental effects for mitochondrial function and partakes in the promotion of cardiac arrhythmias [13]. As AF is associated with increased cytosolic Ca^2+^ [14], and mitochondria buffer cytosolic Ca^2+^ overload [15], it is quite possible that mitochondrial function is compromised. Therefore, in this study we examined the role of mitochondrial function in AF by measuring mitochondrial calcium transients, ATP production, stress chaperones, membrane potential, respiration and morphology in experimental models of AF remodeling and in AF patients. Here, we report that tachypacing of HL-1 atrial cardiomyocytes induces mitochondrial Ca^2+^-handling changes, resulting in reduction in mitochondrial membrane potential and respiration and finally disruption of the network. Furthermore, prevention of mitochondrial Ca^2+^-handling changes, through partial blocking or downregulation of the mitochondrial calcium uniporter (MCU), or improving mitochondrial bioenergetics, by the polypeptide SS31, results in conservation of mitochondrial function and attenuation of AF remodeling. Together, these results identify mitochondria as a novel potential therapeutic target in AF.

## 2. Materials and Methods

### 2.1. HL-1 Cardiomyocyte Cell Culture and Tachypacing

HL-1 atrial cardiomyocytes, derived from adult mouse atria, were obtained from Dr. William Claycomb ([16], Louisiana State University, New Orleans, LA, USA) and maintained in complete Claycomb medium (Sigma, Zwijndrecht, The Netherlands) supplemented with 10% FBS (PAA Laboratories GmbH, Austria), 100 U/mL penicillin (Gibco, Landsmeer, The Netherlands), 100 µg/mL streptomycin (Gibco), 4 mM l-glutamine (Gibco), 0.3 mM l-ascorbic acid (Sigma) and 100 µM norepinephrine (Sigma). The cardiomyocytes were cultured on cell culture plastics or on glass coverslips coated with 0.02% gelatin (Sigma) and were grown at 37 °C in 5% CO_2_.

HL-1 atrial cardiomyocytes were subjected to normal pacing (1 Hz) or tachypacing (6 Hz), 40 V and 20 ms pulses by utilizing the C-Pace100^TM^-culture pacer (IonOptix Corporation, Amsterdam, The Netherlands).

### 2.2. Transfection and Drug Treatments

HL-1 cardiomyocytes were transiently transfected with pDEST40-MCU-V5-HIS (#31731, Addgene, Watertown, MA, USA) by the use of Lipofectamine 2000 (Life Technologies, Bleiswijk, The Netherlands). MCU reduction of 40% and 20% (60% and 80% still present) was accomplished by transiently transfecting the cardiomyocytes with different concentrations Mission esiRNA (EMU213891, Sigma) by the use of Lipofectamine RNAiMAX (Life Technologies), 1 µg and 150 ng, respectively. Ru360 (Millipore, Amsterdam, The Netherlands), mdivi-1 (Sigma), mitoTEMPO (Santa Cruz Biotechnology, Dallas, TX, USA) and SS31 (d-Arg-Dmt-Lys-Phe-NH_2_, prepared by Genscript, Piscataway, NJ, USA) were dissolved according to manufacturer’s instructions. HL-1 atrial cardiomyocytes were treated with Ru360 30 min. prior to pacing, mdivi-1 40 min. prior to pacing and mitoTEMPO and SS31 1 h prior to pacing.

### 2.3. Calcium Transient Measurements

To measure mitochondrial calcium transient (CaT_mito_) amplitudes, HL-1 cardiomyocytes were incubated for 30 min. with 5 µM of Rhod-2 AM ([17], Abcam, Cambridge, UK) at 37 °C in DMEM (Gibco), which provides an indication of changes in mitochondrial Ca^2+^, followed by three times washing with DMEM. To measure cytosolic calcium transient (CaT_cyto_) amplitudes, cardiomyocytes were incubated for 30 min. with 2 µM of the Ca^2+^-sensitive dye Fluo-4-AM (Life Technologies) at 37 °C in DMEM, followed by three times washing with DMEM. Rhod-2 AM-loaded cardiomyocytes were excited by a 600 nm laser with emission at 605 nm and Fluo-4-AM-loaded cardiomyocytes were excited by a 488 nm laser with emission at 500-550 nm. CaT_mito_ and CaT_cyto_ amplitudes were recorded with the Myocyte Calcium and Contractility System (IonOptix Corporation, city, state abbrev. (If USA or Canada), country). The live recording of the CaT amplitudes was performed at 1 Hz stimulation (normal pacing) at 37 °C. The relative value of fluorescent signals was determined utilizing the following calculation: Fcal = F1/F0, where F1 is the fluorescent dye signal at any given time and F0 is the fluorescent signal at rest. Mean values and SEM from each experimental condition were based on 7 consecutive CaTs in at least 25 cardiomyocytes.

### 2.4. Measurements of Mmitochondrial Dysfunction

ATP measurements, mitochondrial morphology analysis, mitochondrial membrane potential analysis and mitochondrial oxygen consumption rate were performed before and after tachypacing.

#### 2.4.1. ATP Measurements

HL-1 atrial cardiomyocytes were lysed and homogenized in 1/20 part 1.5% trichloroacetic acid after which 1 part 1 M Tris-buffer (pH 8.0), supplemented with 1 mM sodium fluoride (NaF), was added, according to the protocol of Promega (ENLITEN ATP assay system bioluminescence detection for ATP measurement – instructions for the use of product FF2000). Human atrial appendage tissue was lysed in 1 part TE-saturated phenol (10 mM Tris-HCl and 1 mM EDTA). After homogenization, 1/10 part chloroform and 1/13 part milliQ was added to ½ part of the homogenized product. The mixture was centrifuged at 10.000× g for 5 min. at 4 °C. The supernatant was diluted 1000× with milliQ and used for ATP measurement [18]. Cellular ATP levels were measured utilizing the ATP Bioluminescence Assay Kit CLSII (Roche, Almere, The Netherlands), according to manufacturer’s instructions. In short, homogenized samples and luciferase reagent (supplied) were added at a 1:1 ratio into a white 96-well plate and ATP levels were measured by bioluminescence using the Mithras LB 940 Multimode Microplate Reader (Berthold Technologies, Bad Wildbad, Germany).

#### 2.4.2. Reactive Oxygen Species (ROS) Measurements

HL-1 atrial cardiomyocytes were lysed in radioimmunoprecipitation (RIPA) assay buffer and the oxidation status of proteins was determined by utilizing the OxyBlot Protein Oxidation Detection Kit (Millipore), according to manufacturer’s instructions. Lipid peroxidation was measured by MDA expression by utilizing the OxiSelect TBARS Assay Kit (Cell Biolabs, San Diego, CA, USA), according to manufacturer’s instructions.

#### 2.4.3. Mitochondrial Morphology Analysis

HL-1 cardiomyocytes were incubated with 100 nM Mitotracker Deep Red (Life Technologies) in DMEM (Gibco) for 30 min. at 37 °C, followed by 2× washing with DMEM and 2× washing with phosphate buffered saline (PBS). Cardiomyocytes were fixated with 4% formaldehyde (Klinipath, Duiven, The Netherlands) for 15 min. at 37 °C, followed by washing 2× with PBS and mounting with Vectashield (Vector Laboratories, Burlingame, CA, USA). Images were obtained by the Zeiss Axiovert 200 M Marianas^TM^ digital imaging inverted microscope system, utilizing a 16-bit cooled charge-coupled device camera (Cooke SensiCam SVGA, Cooke Co., Auburn Hills, MI, USA) with the CY5 filter block and a 63×-oil objective and Slidebook^TM^ (Intelligent Imaging Innovations Inc., Denver, CO, USA) to control hardware and view images. Ten random fields containing at least 10 cardiomyocytes were selected and the mitochondrial morphology per single cardiomyocyte was scored as tubular, intermediate or fragmented by an investigator blinded for the conditions. The network was scored as depicted and described in [19]. Briefly, the network was scored as tubular when it appeared as long, intertwining tubules; as intermediate when the tubules were at least 30% shorter and also showed dots (single mitochondria) in between; and as fragmented when >70% of the network were dots instead of tubules. The amount of tubular, intermediate or fragmented cardiomyocytes were expressed as percentage of the total cardiomyocytes to show the distribution of tubular, intermediate or fragmented mitochondrial morphology between conditions.

#### 2.4.4. Mitochondrial Membrane Potential Analysis

Mitochondrial membrane potential was determined by incubating the HL-1 atrial cardiomyocytes with 100 nM TMRE (ab113852, Abcam) and 100 nM Mitotracker Deep Red (Life Technologies) in DMEM for 20 min. at 37 °C, followed by 1× washing with DMEM, 1× washing with PBS and adding complete Claycomb medium. Live images were obtained by the Zeiss Axiovert 200 M Marianas^TM^ digital imaging inverted microscope system, utilizing a 16-bit cooled charge-coupled device camera (Cooke SensiCam SVGA) with CY3 and CY5 filter blocks and a 63×-oil objective and Slidebook^TM^ (Intelligent Imaging Innovations Inc.) to control hardware and view images. Ten random fields containing at least 10 cardiomyocytes were recorded and mitochondrial membrane potential was analyzed utilizing ImageJ software (v1.49, NIH, Washington, DC, USA). For analysis, the grey intensity and cell area were measured for each separate cardiomyocyte for both TMRE and Mitotracker Deep Red, of which background grey intensity was subtracted. TMRE values were divided by Mitotracker Deep Red values and adjusted for cell area.

#### 2.4.5. Measurement of Mitochondrial Oxygen Consumption Rate

Mitochondrial oxygen consumption rate (OCR) was measured with the Seahorse XFe96 Extracellular Flux Analyzer (Agilent, Santa Clara, CA, USA), utilizing the Seahorse XF Mito Stress Test kit. After tachypacing, HL-1 cardiomyocytes were trypsinized and 3.5 × 10^4^ cardiomyocytes were plated per well in 0.02% gelatin-coated Seahorse XFe96-well cell culture microplates (Agilent) and incubated at 37 °C and 5% CO_2_ for at least 20 h. One hour before the measurement, medium was removed and replaced by DMEM containing 25 mM glucose (Sigma), 1 mM sodium pyruvate (Lonza, Geleen, The Netherlands) and 2 mM l-glutamine (Life Technologies) and cardiomyocytes were incubated in a non-CO_2_ incubator at 37 °C. Mitochondrial respiration was measured before (routine) and after 1.5 µM oligomycin (leak respiration), 0.3 µM FCCP (maximal uncoupled respiration) and 2.5 µM antimycin A and 1.25 µM rotenone (residual respiration). Experiments were performed with 8–10 wells per condition and subsequently averaged. OCR values were adjusted for residual respiration and cell count.

### 2.5. Protein Extraction and Western Blot Analysis

HL-1 atrial cardiomyocytes were lysed in RIPA assay buffer. Human atrial appendage tissue samples were lysed in sample buffer (15% glycerol; 1% SDS; 12,5% 0.5 M Tris, pH 6.8; 2% bromophenol-blue solution). For Western blot analysis, equal amounts of protein lysates were separated on SDS-PAGE 4–20% mini-PROTEAN TGX or 4–20% Criterion TGX precast gels (Bio-Rad, Lunteren, The Netherlands) and transferred to nitrocellulose membranes (Bio-Rad). Subsequently, membranes were incubated with primary antibodies. Signals were detected by the Amersham ECL prime Western blotting detection reagent (GE Healthcare Life Sciences, Hoevelaken, The Netherlands) utilizing the Amersham Imager 600 (GE Healthcare Life Sciences) and quantified by densitometry (ImageQuantTL, GE Healthcare Life Sciences). The following primary antibodies were used: Anti-HSP60 (ADI-SPA-805, Enzo Life Sciences, Farmingdale, NY, USA), anti-TOM20 (MCA4300Z, Bio-Rad), anti-MCU (14997S Cell Signalling Technology, Leiden, The Netherlands), OXPHOS Antibody Cocktail (MS604, Abcam) and anti-GAPDH (10R-G109a, Fitzgerald, Acton, MA, USA). Horseradish peroxidase-conjugated anti-mouse or anti-rabbit (Dako, Denmark) were used as secondary antibodies, depending on the species origin of the primary antibody.

### 2.6. Quantitative RT-PCR

Total RNA was isolated from HL-1 atrial cardiomyocytes using the Nucleospin RNA isolation kit (Macherey-Nagel, Landsmeer, The Netherlands). First strand cDNA was generated by utilizing the iScript cDNA synthesis kit (Bio-Rad) and subsequently used as a template for quantitative real-time PCR. Relative changes in transcription level were determined utilizing the CFX384 Real-time system C1000 Thermocycler (Bio-Rad) in combination with SYBR Green Supermix (Bio-Rad). mRNA levels were expressed in relative units on the basis of a standard curve (serial dilutions of a calibrator cDNA mixture) and adjusted for *GAPDH* levels. PCR efficiencies for all primers were between 90–110%. Primer pairs used are the following: *HSP60* fw: TGACTTTGCAACAGTCACCC and rv: GCTGTAGCTGTTACAATGGGG, *HSP10* fw: CTCCAACTTTCACACT-GACAGG and rv: GCCGAAACTGTAACCAAAGG, *MCU* fw: ACTGTTCTGGGGACAGGATG and rv: ACCT-CTGCAAGGCACCTAGA and *GAPDH* fw: CATCAAGAAGGTGGTGAAGC and rv: ACCACCCTGTTGCTGTAG.

Total DNA was isolated from HL-1 cardiomyocytes utilizing the Nucleospin Tissue kit (Macherey-Nagel), according to manufacturer’s instructions. Isolated DNA was used to determine mitochondrial DNA levels utilizing the CFX384 Real-time system C1000 Thermocycler (Bio-Rad) in combination with SYBR Green Supermix (Bio-Rad). Mitochondrial DNA levels were adjusted for nuclear DNA levels and analyzed using the ΔC_T_ method [20]. Primer pairs used are the following: *COX1* fw: GCCCCAGATATAGCATTCCC and rv: GTTCATCCTGTTCCTGCTCC, *18S rRNA*: TAGAGGGACAAGTGGCGTTC and rv: CGCTGAGCCAGTCAGTGT.

### 2.7. Drosophila Melanogaster Heart Rate Measurements

*Drosophila melanogaster* heart rate measurements were performed with the *w^1118^* strain (Genetic Services Inc., Sudbury, MA, USA). All flies were maintained at 25 °C on standard medium. Five Male and five female *Drosophilas* were added per tube and after 4 days adult flies were removed and drugs (Ru360 or SS31) were added to the medium, which contained fly embryos. Controls were treated with the vehicle demineralized water. Larvae consumed the drug or vehicle during 3 days, after which transparent prepupae were collected and placed on a 1% agarose gel in PBS and subjected to tachypacing (5 Hz for 20 min., 20 V and 5 ms pulses) with a C-Pace100^TM^ culture pacer (IonOptix Corporation). High-speed movies of spontaneous heart wall contractions in prepupae were recorded before and after tachypacing for 3 × 10 s. at a rate of 100 frames per second by using a BlueFOX3 digital camera on a Leica DM IL LED microscope with a 10×-objective. Heart rate was manually analyzed with ImageJ.

### 2.8. Patient Material

Before surgery, patient characteristics were assessed (Table 1). Definitions for AF stages are as follows: Paroxysmal (episodes last less than one week), persistent (episodes last longer than one week) and longstanding persistent (episodes last more than one year). Duration of AF is based on documented information stating the diagnosis of AF. Right and left atrial appendages were obtained from patients with AF and patients in sinus rhythm (SR) undergoing cardiac surgery and included in the Halt and Reverse study (MEC 2014-393) [21]. All AF and SR patients have underlying heart disease. After excision, the atrial appendages were immediately snap-frozen in liquid nitrogen and stored at −80 °C. The study conforms to the principles of the Declaration of Helsinki. The institutional board of the Erasmus Medical Center, Rotterdam, the Netherlands approved the study and patients gave written informed consent.

### 2.9. Statistical Analysis

Results are expressed as mean ± SEM of 3–5 independent experiments for Western blot, RT-PCR, ROS, ATP and OCR measurements, 3-5 independent experiments containing at least 25 or 100 cardiomyocytes per experiment for CaT and mitochondrial morphology/membrane potential measurements, respectively and 15–20 *Drosophila* prepupae for heart rate analyses. Individual group-mean differences were evaluated with a Student’s t-test and the Benjamini-Hochberg procedure was used to adjust for multiple testing. All *p*-values were two-sided and a value of *p* ≤ 0.05 was considered statistically significant. SPSS version 22 was used for all statistical evaluations.

## 3. Results

### 3.1. Tachypacing Induces Mitochondrial Dysfunction and Stress

It has been recognized that AF is associated with increased cytosolic Ca^2+^ levels and mitochondria are known to buffer Ca^2+^ [14,15]. Therefore, we examined the effect of tachypacing on mitochondrial function, by first studying mitochondrial Ca^2+^-handling, by means of measuring mitochondrial calcium transient (CaT_mito_) amplitudes. Initially, tachypacing increased CaT_mito_ amplitude (Appendix A). However, after 6 h tachypacing CaT_mito_ were significantly reduced in amplitude compared to normal-paced HL-1 cardiomyocytes (Figure 1A and Appendix A). These results are in concordance with previous findings that extended periods of mitochondrial Ca^2+^ buffering reduce mitochondrial Ca^2+^ uptake [15]. As the CaT_mito_ amplitude is dependent on the mitochondrial membrane potential (ΔΨ_mito_) [15], ΔΨ_mito_ was measured by using the fluorescent probe TMRE, which is readily sequestered in polarized mitochondria. In line with CaT_mito_ amplitude findings, tachypacing significantly and progressively reduced ΔΨ_mito_ (Figure 1B). Next, we investigated whether tachypacing induces mitochondrial stress by measuring the levels of the mitochondrial stress-related chaperones *HSP60* and *HSP10*, which are essential for mitochondrial function as they are important players in synthesis, folding, transport and maintenance of mitochondrial proteins [22]. Tachypacing resulted in a significant and progressive upregulation of *HSP60* and *HSP10* mRNA, suggesting the presence of mitochondrial stress (Figure 1C,D).

To test whether tachypacing-induced CaT_mito_ amplitude loss is associated with changes in mitochondrial function, mitochondrial respiration was examined. Tachypacing resulted in proton leak and reduction in ATP production (Figure 1E–G, Appendix A, [23]). Furthermore, tachypacing initially increased the maximal respiration, which returned to control levels at longer duration of tachypacing (Figure 1H). The decline in maximal respiration from 6 to 8 h of tachypacing is in concordance with the decline of mitochondrial functions at 8 h of tachypacing, implying that the increased maximal respiration at 6 h of tachypacing is a result of the cardiomyocytes trying cope with the high activation rate. This suggests that mitochondria are able to respond to short-term tachypacing by compensation mechanisms reflected by the increased maximal respiration and subsequent increased CaT_mito_ amplitude.

Next to functional changes, we studied whether tachypacing results in mitochondrial structural changes. Hereto, we examined whether tachypacing causes alterations in mitochondrial mass. Tachypacing did not affect protein abundance of electron transport system (ETS) complexes I, II, III and V (Figure 1I). Similarly, no differences in the amount of mitochondrial DNA or TOM20 levels were observed after tachypacing (Figure 1J,K), suggesting that tachypacing did not affect mitochondrial mass. In addition, we examined the morphology of the mitochondrial network. Tachypacing induced a time-dependent transition from a tubular to a fragmented network (Figure 1L and Appendix A), indicating a compromised mitochondrial bioenergetic function by reducing the optimal tubular configuration of mitochondria [24].

Together, these data demonstrate that tachypacing progressively impairs mitochondrial function in HL-1 cardiomyocytes, by impairment of mitochondrial Ca^2+^-handling, increasing mitochondrial stress, loss of ΔΨ_mito_, reduction in respiration and ATP production and finally fragmentation of the mitochondrial network. Total mitochondrial complex protein content or mass remained, however, intact.

### 3.2. Markers of Mitochondrial Dysfunction are Present in AF Patients

To extend our findings to human AF, we examined mitochondrial dysfunction in left and/or right atrial appendages (LAA and RAA, respectively) from patients with paroxysmal (PAF), persistent (PeAF) and longstanding persistent (LSPeAF) AF and patients in sinus rhythm (SR). Given the small amount of LAA tissue samples from SR patients, we only were able to measure ATP levels. Cellular ATP levels were significantly increased in LAA compared to RAA. ATP levels in LAA of PAF and PeAF patients were significantly increased compared to LAA of SR patients, but reduced again in LSPeAF patients (Figure 2A). This finding suggests an initial compensatory mechanism to sustain the high heart rate of AF, which gets exhausted when AF persists for a longer period of time. Mitochondrial dysfunction is further evidenced in PeAF and LSPeAF by increased protein expression of HSP60 in both RAA and LAA (Figure 2B), while there is no change in expression of TOM20 (Figure 2C) and of respiratory system complexes I, III, IV and V (Figure 2D). Due to antibody specificity, complexes I, III IV and V were detected in patients compared to complexes I, II, III and V in HL-1 cardiomyocytes. In addition, upon electron microscopic examination, patients in SR have mitochondria localized along the entire length of intact sarcomeres, whereas PeAF shows fragmented and dispersed mitochondria and degraded sarcomeres (denoting myolysis, Figure 2E) [25].

These results indicate that mitochondrial dysfunction, due to mitochondrial fragmentation and not changes in mitochondrial mass, is found in an *in vitro* model of AF and is also present in AF patients.

### 3.3. Tachypacing-Induced CaT_mito_ Loss Was Prevented by the Mitochondrial Ca^2+^ Uniporter Inhibitor Ru360 and Peptide SS31

As tachypacing induces impairment of mitochondrial function, we next tried to uncover by which mechanism(s) mitochondria are impaired. To this end, the effects of four compounds, targeting mitochondrial function via different pathways, were explored in HL-1 cardiomyocytes. These compounds were: mdivi-1, an inhibitor of mitochondrial fragmentation [26]; mitoTEMPO, a mitochondrial antioxidant [27]; Ru360, an inhibitor of the mitochondrial calcium uniporter (MCU) [28] and SS31, a peptide that conserves the ETS and thereby mitochondrial bioenergetics [29]. The effects of these compounds to counteract tachypacing-induced loss of CaT_mito_ amplitude was examined after 6 h of tachypacing, as at this time point CaT_mito_ amplitudes were significantly reduced (Figure 1A and Appendix A). Both mdivi-1 and mitoTEMPO did not protect against tachypacing-induced CaT_mito_ amplitude reduction at any concentration applied (Figure 3A,B). Treatment with mitoTEMPO even appears to exacerbate the phenotype of tachypacing, which may be due to such a substantial decrease in mitochondrial ROS levels that it interferes with the physiological function of ROS. In contrast, Ru360 significantly protected at higher concentrations against CaT_mito_ amplitude reductions due to tachypacing (Figure 3C). SS31, a compound used in clinical trials, such as for heart failure and mitochondrial myopathy, showed protection against tachypacing-induced CaT_mito_ amplitude reduction at the highest concentration applied (Figure 3D). These results suggest that Ru360 and SS31, targeting different mitochondrial mechanisms by MCU inhibition and mitochondrial ETS and bioenergetics conservation, respectively, protect against loss of tachypacing-induced CaT_mito_ amplitudes.

### 3.4. Ru360 Protects against Tachypacing-Induced Mitochondrial Dysfunction

Tachypacing-induced CaT_mito_ amplitude loss was protected by administration of 5 µM Ru360 (Figure 3C), a compound which specifically blocks Ca^2+^-influx through the MCU [28]. We explored whether Ru360 treatment also ameliorated mitochondrial dysfunction and stress. Ru360 treatment normalized cellular ATP levels (Figure 4A and Appendix A) and protected the mitochondrial network from tachypacing-induced fragmentation (Figure 4B, Appendix A). However, Ru360 treatment did not normalize transcription levels of *HSP60* and *HSP10* after tachypacing compared to normal-paced cardiomyocytes (Figure 4C,D and Appendix A), although the transcription levels were marginally decreased, which may imply that mitochondrial stress is induced before the aberrant mitochondrial Ca^2+^-handling.

To substantiate effectiveness of Ru360 in another model, we treated tachypaced *Drosophila melanogaster* with different concentrations of Ru360. Comparable to findings in tachypaced HL-1 cardiomyocytes, Ru360 conferred protection against tachypacing-induced decrease of heart wall contraction (heart rate) in *Drosophila* (Figure 4E and Appendix A). The optimal concentration of Ru360 needed, 50 µM in *Drosophila* as opposed to 5 µM in HL-1 cardiomyocytes, corresponds well with prior experiments in which concentrations needed to confer protection in *Drosophila* are consistently 10x higher than in HL-1 cardiomyocytes [30].

Together, these results show that inhibition of the MCU-mediated Ca^2+^-influx into the mitochondria by Ru360 protects against mitochondrial dysfunction and stress and preserves cellular function in tachypaced cardiomyocytes.

### 3.5. The MCU may Mediate Tachypacing-Induced Mitochondrial Changes

To determine whether the MCU plays an important role in tachypacing-induced mitochondrial dysfunction, we studied the role of the MCU in more detail. Hereto, we first examined its protein level, which was not affected by tachypacing (Figure 5A, Appendix A). Next, we manipulated MCU levels by overexpression or knockdown. Overexpression of the MCU did not affect tachypacing-induced loss of CaT_mito_ amplitudes (Figure 5B, Appendix A). In contrast, reducing *MCU* expression, via siRNA treatment, by 20% (Appendix A) protected against tachypacing-induced CaT_mito_ amplitude loss (Figure 5C, Appendix A) and significantly increased the CaT_mito_ amplitudes in normal-paced cardiomyocytes (Appendix A). Interestingly, a 40% reduction of *MCU* expression (Appendix A) lowered CaT_mito_ amplitudes in normal-paced cardiomyocytes and conferred no protection in tachypaced cardiomyocytes (Figure 5C, Appendix A). Our results suggest that a modest reduction in the MCU abundance does not affect normal mitochondrial Ca^2+^-handling and may preserve adequate MCU function and is, therefore, beneficial to counteract further tachypacing-induced mitochondrial dysfunction. However, a larger reduction in MCU level seems detrimental, likely due to an impairment of physiological mitochondrial Ca^2+^-influx, which is already observed under baseline conditions.

### 3.6. SS31 Protects against Tachypacing-Induced Mitochondrial Dysfunction and Stress

As Ru360 is, at the moment, only used in a pre-clinical setting, we explored whether SS31, a compound already used in clinical trials (trade name elamipretide), also ameliorated mitochondrial dysfunction and stress. SS31 showed a significant protection against tachypacing-induced CaT_mito_ amplitude reduction at the highest concentration applied (100 nM, Figure 3D). SS31 treatment normalized cellular ATP levels (Figure 6A, Appendix A) and partially attenuated contractile dysfunction, as shown by attenuation of cytosolic CaT loss after tachypacing (Figure 6B, Appendix A). In addition, SS31 treatment attenuated tachypacing-induced loss of ΔΨ_mito_ (Figure 6C, Appendix A) and reduced *HSP60* and *HSP10* transcription levels (Figure 6D,E and Appendix A). Next, we examined whether SS31 treatment protected against the loss of mitochondrial respiration. We found that SS31 treatment significantly increases routine respiration and ATP production after 8 h tachypacing, but did not prevent tachypacing-induced increase in leak respiration or maximal respiration (Figure 6F–I, Appendix A). Furthermore, we determined mitochondrial morphology and show that SS31 protects the network from tachypacing-induced fragmentation (Figure 6J, Appendix A). Finally, we explored the effect of different concentrations (1, 5 and 10 µM) of SS31 in tachypaced *Drosophila melanogaster*. In contrast to findings in tachypaced HL-1 cardiomyocytes, SS31 did not confer protection against tachypacing-induced decrease in heart wall contractions in *Drosophila* (Figure 6K, Appendix A).

Together, these results suggest that restoring mitochondrial bioenergetics and improving ETS function by SS31 protects against mitochondrial dysfunction and stress and preserves cellular function in tachypaced HL-1 cardiomyocytes.

## 4. Discussion

In the current study, we show tachypacing to progressively induce mitochondrial dysfunction, evidenced by impairment of mitochondrial Ca^2+^-handling, upregulation of mitochondrial stress chaperones, collapse of the mitochondrial membrane potential, decrease of mitochondrial respiration and ATP levels and fragmentation of the mitochondrial network. Our data suggests that enhanced Ca^2+^-influx through the MCU might be an important pathophysiological mechanism, consequently impairing mitochondrial calcium transients. In line, mitochondrial changes, including aberrant ATP levels and increased HSP60 expression, are present in atrial tissue of AF patients, which also show structural remodeling (myolysis) and fragmented and dispersed mitochondrial localization. Treatment with Ru360, an inhibitor of the MCU, or modest MCU downregulation prevented tachypacing-induced detrimental mitochondrial changes. Furthermore, Ru360 treatment protected against contractile dysfunction in a *Drosophila melanogaster* model for AF. In addition, SS31, a compound used in clinical trials to improve ETS function and restore mitochondrial bioenergetics, protected against tachypacing-induced detrimental mitochondrial changes in HL-1 cardiomyocytes. Together, these results suggest that targeting mitochondria may represent a novel therapeutic strategy to counteract AF-induced mitochondrial dysfunction.

### 4.1. Mitochondrial Dysfunction and Its Implication in Cardiac Disease

The findings observed in tachypaced HL-1 cardiomyocytes and *Drosophila* and clinical AF are in line with other cardiac diseases. Multiple cardiac diseases, including heart failure, myocardial infarction, ischemic heart disease, dilated cardiomyopathy, diabetic cardiomyopathy and hypertension-induced cardiomyopathy, are associated with mitochondrial dysfunction, including mitochondrial ATP depletion, decreased mitochondrial respiration and membrane potential and aberrant mitochondrial morphology [31,32]. Aberrant mitochondrial morphology includes mitochondrial fragmentation and swelling, disorganized and/or dismantled cristae, smaller mitochondria and/or reduced amount of mitochondria [31]. Both the mitochondrial fragmentation [33] and destroyed cristae [34] result in impairment of mitochondrial respiration, which compromise ATP production and, consequently, cardiac contraction. Moreover, we observed reduced CaT_mito_, which is in concordance with previous findings that extended periods of mitochondrial Ca^2+^-buffering reduce mitochondrial Ca^2+^ uptake, which, in turn, negatively affects mitochondrial respiration [15]. Our findings of ATP levels in patients suggest an initial compensatory mechanism to sustain the high heart rate of AF, which gets exhausted when AF persists for a longer period of time. The observed differences between ATP levels in atrial tissue of patients is likely due to heterogeneity of the patient population. At this moment, it’s the chicken or the egg question whether AF leads to mitochondrial dysfunction or whether mitochondrial dysfunction drives AF. However, it is believed that structural changes are already present before AF onset and that AF accelerates further changes [35,36]. We also show that changes are more prominent in the LAA of AF patients, which is in line with evidence that AF affects RA and LA differently [37,38]. Although it is known that AF affects RA and LA differently, the specific mechanisms are still unknown, but may be caused by the dilatation of the LA (which is not present in the RA).

As mitochondria comprise approximately 30% [39] of the cardiomyocyte volume and account for 90% [40] of the provided cardiac contraction energy, mitochondrial dysfunction is detrimental for the heart. This is exemplified by the vast amount of cardiac diseases caused or worsened by mitochondrial dysfunction. Likewise, in this study we provide evidence that mitochondrial dysfunction may also underlie AF progression.

### 4.2. MCU-Mediated Mitochondrial Dysfunction and Stress in AF

Our data indicate that mitochondrial dysfunction and stress in an *in vitro* model of AF-related remodeling may be caused by MCU-mediated Ca^2+^-influx. The MCU regulates the Ca^2+^-influx through the inner mitochondrial membrane and is important for maintenance of the mitochondrial Ca^2+^ homeostasis, which, in turn, is essential for cellular physiology [41]. The MCU is specifically blocked by Ru360 [28], which protects against tachypacing-induced mitochondrial dysfunction in the current study, possibly by decreasing mitochondrial Ca^2+^-influx. Such view is consistent with observations on the key role of MCU in heart failure [42] and findings that Ru360 protects against ischemia-reperfusion injury in *in vitro* [43] and *in vivo* [44] models. Additionally, next to its inhibition of Ca^2+^-influx, Ru360 may also stimulate mitochondrial Ca^2+^-efflux, as observed for its analog ruthenium red in rat [45].

The optimal treatment dose of Ru360 used in this study, 5 µM, does not confer a complete block of the MCU. A complete block is accomplished by treatment with 10 µM Ru360 (as stated by the manufacturer), a concentration that showed fewer protective effects in the current study. Similarly, we found *MCU* siRNA treatment reducing *MCU* expression by 20% to protect cardiomyocytes against tachypacing-induced CaT_mito_ loss, in contrast to a 40% reduction of *MCU* expression, which showed no protection and even revealed CaT_mito_ reduction under normal-paced conditions. The latter is in agreement with other studies [46], in which a reduction in *MCU* expression of 60% or more was not protective in rat pancreatic islets. Whereas a modest reduction, as we show in the current study, preserved adequate MCU function.

In line, experiments in MCU^−/−^ mice support that complete or almost complete downregulation of MCU has detrimental effects [47]. The role of MCU in normal cardiomyocyte physiology is in accordance with a study showing increased cytosolic Ca^2+^ oscillations in *MCU* siRNA-transfected cardiomyocytes [48]. Cytosolic Ca^2+^ levels influences mitochondrial morphology [49] and cytosolic Ca^2+^ overload leads to mitochondrial fragmentation [44]. In turn, mitochondrial fragmentation impairs the function of respiratory chain system [24] and, consequently, results in decreased ATP production. Studies revealed that a reduction in ATP production increases the open state probability of sarcolemmal K_ATP_ channels, and thereby reduces the length of the effective refractory period and action potential duration, and promoting development of arrhythmias [50], such as AF [51]. Together, these data suggest that a modest decrease in MCU level inhibits tachypacing-induced impairment of mitochondrial Ca^2+^-handling and thereby protects against further detrimental effects on cardiomyocyte function.

### 4.3. Amelioration of Tachypacing-Induced Mitochondrial Dysfunction by SS31

Our data also indicate SS31 (trade name elamipretide) as a promising therapeutic compound in AF. SS31 acts as a ROS scavenger and cardiolipin stabilizer, thereby improving ETS function and restoring mitochondrial and cellular bioenergetics [29]. SS31 is a tetra-peptide that belongs to the Szeto-Schiller (SS) peptides, and targets specifically the inner mitochondrial membrane (IMM), where its maximal concentration (5000-fold) is reached within 2 min. The IMM is difficult to target by drugs as it is highly impermeable to molecules due to its high concentration of mitochondrial proteins and cardiolipin. Cardiolipin is an IMM-specific phospholipid that is required for optimal ETS function and cristae formation. Cardiolipin organizes the ETS on the cristae as supercomplexes to optimize mitochondrial bioenergetics [52]. SS31 binds to cardiolipin, resulting in improved ETS function and ATP synthesis [53]. In addition, SS31 scavenges ROS and inhibits mitochondrial swelling and opening of the mitochondrial permeability transition pore, thereby preventing apoptosis [29].

We show that SS31 treatment improves several mitochondrial parameters after tachypacing, including ATP levels, mitochondrial membrane potential and mitochondrial morphology. These beneficial effects are also found for SS31 treatment in *in vitro* and *in vivo* neurodegenerative [54] and other cardiac diseases, including heart failure and myocardial infarction [55,56].

SS31 treatment did not confer protection in tachypaced *Drosophila*, which may be due to the short half-life of SS31 (0.8–2 h) [52,57]. In *Drosophila*, SS31 was added to the food for 3 days after which prepupae were collected and subjected to tachypacing. Therefore, it is highly likely that SS31 is not functional at the moment of heart function measurement. However, injection of SS31 directly in the thorax of adult *Drosophila* was shown to confer protection against trauma-induced thoracic injury [58].

With respect to clinical dose, half-life and side effects, much may be learned from outcomes of ongoing human trials with elamipretide in several diseases. A small phase I study with elamipretide in heart failure patients showed promising results, as a single high infusion dose was well tolerated in patients [59]. In addition, SS31 treatment resulted in less frequent and severe arrhythmias after myocardial infarction [56]. Previously, we showed that ER stress-associated autophagy importantly contributed to proteostasis derailment in AF and AF-induced structural remodeling [9]. Interestingly, SS31 was able to attenuate increased autophagic activation in an *in vivo* model of heart failure [60] and attenuated ER stress in an *in vivo* model of major burn injury [61]. In addition, RhoA signaling was inhibited by SS31 treatment [60], which may be beneficial in AF as RhoA activation was found to be detrimental by suppressing the protective heat shock response (HSR), thereby sensitizing cardiomyocytes to proteotoxic stress [62]. Although the *in vitro* applied concentration of SS31 in this study (100 nM) does not confer protection at all measured mitochondrial parameters, a combination of SS31 with another protective experimental compound, such as GGA [63] and/or 4PBA [9] or currently used anti-arrhythmic drugs, may increase the protective effect against AF-induced remodeling in the clinic. Application of these combined compounds in (SS31-conjugated) nanoparticles [64] might be an interesting option to explore in future studies.

### 4.4. Strengths and Limitations

One of the strengths of this study is the utilization of multiple different experimental techniques to determine whether there is mitochondrial dysfunction in AF. All these techniques show in independent experiments similar results, indicating that mitochondrial dysfunction underlies experimental and clinical AF.

Although our *in vitro* HL-1 atrial cardiomyocyte model system, utilized in this study, has limitations, it shares features of adult cardiomyocytes [16]. Although there may be differences, the important advantage of utilizing the HL-1 cardiomyocyte model system is the use of genetic and pharmacological manipulation to confirm specific molecular pathways involved in tachypacing-induced remodeling. In addition, we have repeatedly confirmed the findings in the tachypaced HL-1 model in the tachypaced dog model and clinical human AF [9,10,65]. The advantage of this model is that only 6-8 h of tachypacing is the most optimal condition to dissect molecular pathways involved in clinical AF progression. In this condition, the HL-1 atrial cardiomyocyte model reproduces structural changes as observed in paroxysmal and persistent AF patients [10,66,67].

One limitation of this study is the lack of LAA tissue from SR patients. Because of ethical issues, LAA tissue may only be collected in case of a medical reason. Previous studies in our group ([10,23]) showed that the expression level of proteins is quite similar in LAA and RAA of SR patients, which is also the case in the ATP assay in this study. Therefore, we are confident that protein expression level in LAA of AF patients is indeed increased.

Another issue with the included patients is the insuperable differences in medication regime. Medication may influence mitochondrial function. Since the drug usage within the various stages of AF is comparable, one may argue that the effect on mitochondrial function within the AF groups is due to the arrhythmia and not to variations in medication.

The use of Rhod-2 AM as a mitochondrial Ca^2+^-indicator is controversial. However, when HL-1 cardiomyocytes, simultaneously treated with Rhod-2 AM and Fluo-4 AM, were subjected to short-term tachypacing, a different effect on the CaT amplitudes between both dyes were observed, indicating that calcium fluxes in different cellular compartments were measured, i.e., mitochondria for Rhod-2 AM and cytosol for Fluo-4 AM. Moreover, several other studies successfully utilized Rhod-2 AM as a mitochondrial Ca^2+^-indicator [68,69]. Therefore, we are confident that the measured mitochondrial CaT with Rhod-2 AM reflect the actual calcium fluxes within the mitochondria.

The MCU OE plasmid showed an increased MCU protein expression, however, we did not determine the localization to the mitochondria. Nevertheless, functionally we show that MCU OE and downregulating of *MCU*, by siRNA, has a detrimental and beneficial effect, respectively on CaT_mito_, indicating that MCU modulated mitochondrial calcium.

## 5. Conclusions

The current study implicates that mitochondrial dysfunction is involved in AF promotion and that targeting of mitochondria may represent a novel therapeutic strategy to counteract AF-induced mitochondrial dysfunction.

## Figures and Tables

**Figure 1 cells-08-01202-f001:**
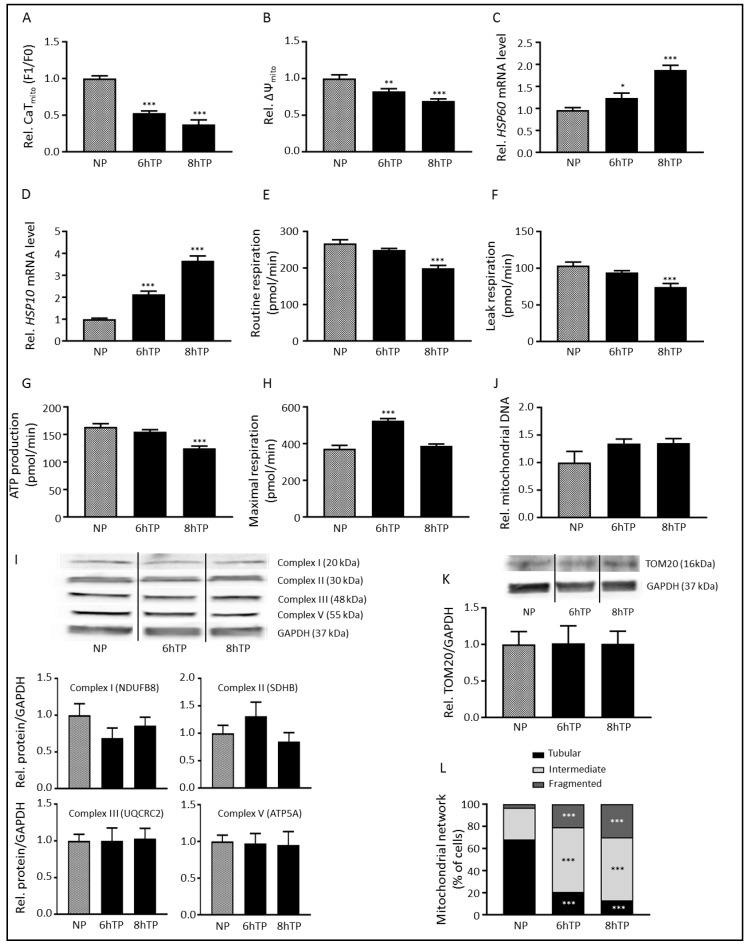
Tachypacing induces mitochondrial dysfunction and stress. (**A**) Quantified data of mitochondrial calcium transient (CaT_mito_) amplitudes of HL-1 atrial cardiomyocytes during normal-pacing (NP) and tachypacing (TP). (**B**) Mitochondrial membrane potential during NP and TP. Quantitative real-time PCR of mitochondrial stress markers (**C**) *HSP60* and (**D**) *HSP10* in response to NP and TP. The oxygen consumption rate showing (**E**) routine respiration, (**F**) leak respiration, (**G**) ATP production and (**H**) maximal respiration during NP and TP. (**I**) Top panel represent Western blot of respiratory chain complexes I, II, III and V. Lower panels reveal quantified data of the respiratory chain complexes normalized for GAPDH. (**J**) No changes in mitochondrial DNA during TP compared to NP. (**K**) Top panel represent Western blot and lower panel reveal quantified data of TOM20 normalized for GAPDH. (**L**) Transition of the mitochondrial network from tubular to fragmented from NP to TP. * *p* < 0.05, ** *p* < 0.01, *** *p* < 0.001 vs. NP.

**Figure 2 cells-08-01202-f002:**
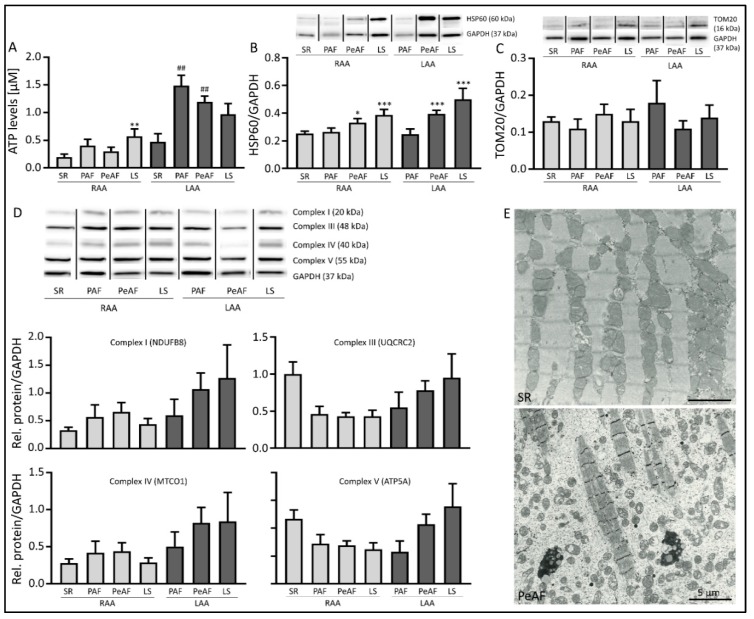
AF patients show mitochondrial dysfunction. (**A**) Cellular ATP levels in right atrial appendage (RAA) or left atrial appendage (LAA) of patients in sinus rhythm (SR), paroxysmal (PAF), persistent (PeAF) or longstanding persistent (LS) AF. (**B**) Top panel represent Western blot of HSP60 and GAPDH, lower panel reveals quantified data of HSP60 normalized for GAPDH. (**C**) Top panel represent Western blot of TOM20 and GAPDH, lower panel reveals quantified data of TOM20 normalized for GAPDH. (**D**) Top panel represent Western blot of respiratory chain complexes I, III, IV and V. Lower panels reveal quantified data of the respiratory chain complexes normalized for GAPDH in RAA of SR and RAA and LAA of PAF, PeAF and LS patients. (**E**) Electron microscopic images of LAA of a patient in SR (top), showing normal sarcomere structure and mitochondrial localization along the sarcomeres, and LAA of PeAF, showing myolysis and dispersed mitochondria. * *p* < 0.05, ** *p* < 0.01, *** *p* < 0.001 vs. SR RAA, ^##^
*p* < 0.01 vs. SR LAA.

**Figure 3 cells-08-01202-f003:**
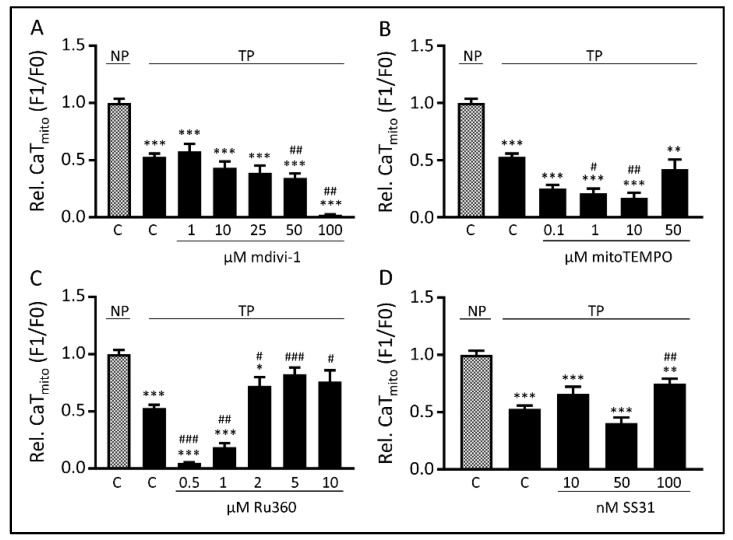
Ru360 and SS31, but not mdivi-1 or mitoTEMPO protect against tachypacing-induced CaT_mito_ amplitude loss. CaT_mito_ amplitude of normal-paced (NP) or 6h tachypaced (TP) with different concentrations of (**A**) mdivi-1, (**B**) mitoTEMPO, (**C**) Ru360 and (**D**) SS31. * *p* < 0.05, ** *p* < 0.01, *** *p* < 0.001 vs. NP C, ^#^
*p* < 0.05, ^##^
*p* <0.01, ^###^
*p* < 0.001 vs. TP C.

**Figure 4 cells-08-01202-f004:**
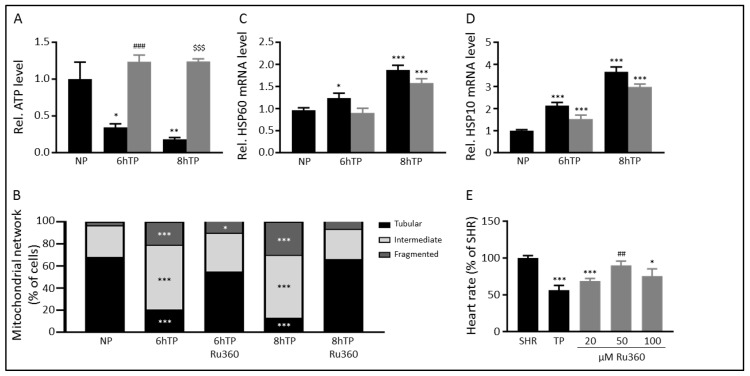
Inhibition of the MCU by Ru360 protects against mitochondrial dysfunction and stress. (**A**) Protection of Ru360 (5 µM) treatment on cellular ATP levels after normal-pacing (NP) or tachypacing (TP). Black bars represent non-treated HL-1 cardiomyocytes, grey bars represent Ru360-treated cardiomyocytes. (**B**) The transition of the mitochondrial network from tubular to fragmented from NP to TP with and without Ru360 treatment. Quantitative real-time PCR of mitochondrial stress markers show no protection of Ru360 on (**C**) *HSP60* and (**D**) *HSP10* transcription levels in response to NP and TP. Black bars represent non-treated HL-1 cardiomyocytes, grey bars represent Ru360-treated cardiomyocytes. (**E**) Quantified data showing heart wall contraction rates (heart rate) before (SHR, spontaneous heart rate) and after tachypacing (TP and grey bars). Black bars represent non-treated *Drosophila*, grey bars represent Ru360-treated *Drosophila*. * *p* < 0.05, ** *p* < 0.01, *** *p* <0.001 vs. NP or SHR, ^##^
*p* < 0.01, ^###^
*p* < 0.001 vs. TP (6 h), ^$$$^
*p* < 0.001 vs. TP (8 h).

**Figure 5 cells-08-01202-f005:**
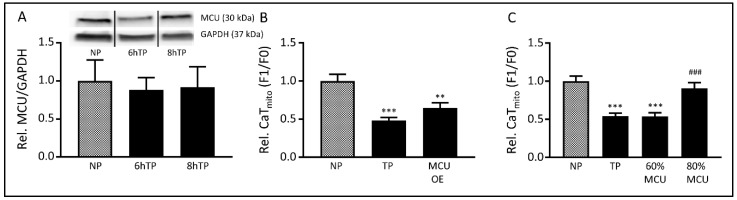
Tachypacing-induced changes are due to MCU expression. (**A**) Top panel represent Western blot of MCU and GAPDH, lower panel reveals quantified data of MCU normalized for GAPDH during normal-pacing (NP) or tachypacing (TP). (**B**) Quantified data of mitochondrial calcium transient (CaT_mito_) amplitudes of HL-1 atrial cardiomyocytes either non-transfected or transfected with MCU, generating MCU overexpression (MCU OE) after NP or 6h tachypacing (black bars). (**C**) Quantified data of CaT_mito_ amplitudes of HL-1 atrial cardiomyocytes either non-transfected or transfected with *MCU* siRNA, generating *MCU* knockdown so that either 60% or 80% or MCU is still present after NP or 6h tachypacing (black bars). ** *p* < 0.01, *** *p* < 0.001 vs. NP, ^###^
*p* < 0.001 vs. TP.

**Figure 6 cells-08-01202-f006:**
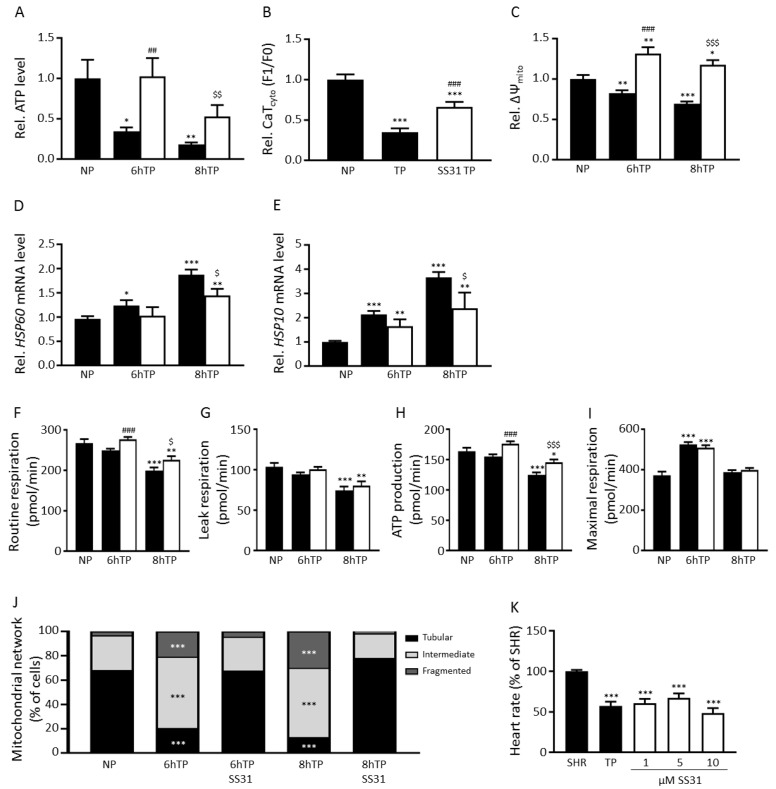
Treatment with SS31 protects against mitochondrial dysfunction and stress. (**A**) Quantified data showing protection of SS31 (100 nM) treatment on cellular ATP levels after normal-pacing (NP) or tachypacing (TP). Black bars represent non-treated HL-1 cardiomyocytes, white bars represent SS31-treated cardiomyocytes. (**B**) Quantified data of mitochondrial calcium transient (CaT_mito_) amplitudes of HL-1 atrial cardiomyocytes after NP or TP. (**C**) Quantified data showing protection of SS31 treatment on mitochondrial membrane potential during NP and TP. Quantitative real-time PCR of mitochondrial stress markers show a slight protection of SS31 after 8h TP on (**D**) *HSP60* and (**E**) *HSP10* transcription levels in response to NP and TP. Black bars represent non-treated HL-1 cardiomyocytes, white bars represent SS31-treated cardiomyocytes. The oxygen consumption rate showing (**F**) routine respiration, (**G**) leak respiration, (**H**) ATP production and (**I**) maximal respiration during NP and TP with and without SS31 treatment. (**J**) The transition of the mitochondrial network from tubular to fragmented from NP to TP with and without SS31 treatment. (**K**) Quantified data showing heart wall contraction rates (heart rate) before (SHR, spontaneous heart rate) and after tachypacing (TP and white bars). Black bars represent non-treated *Drosophila*, white bars represent SS31-treated *Drosophila*. * *p* < 0.05, ** *p* < 0.01, *** *p* < 0.001 vs. NP or SHR, ^##^
*p* < 0.01, ^###^
*p* < 0.001 vs. TP (6 h), ^$^
*p* < 0.05, ^$$^
*p* < 0.01, ^$$$^
*p* < 0.001 vs. TP (8 h).

**Table 1 cells-08-01202-t001:** Demographic and clinical characteristics of patients with atrial fibrillation (AF) and patients in sinus rhythm (SR), used for ATP measurements and Western blot analysis of atrial appendages.

	SR	PAF	PeAF	LSPeAF
N	35	14	24	14
RAA	33	13	22	12
LAA	7	4	19	8
Gender				
Male (N, %)	26 (74.3)	10 (71.4)	15 (62.5)	12 (85.7)
Female (N, %)	9 (25.7)	4 (28.6)	9 (37.5)	2 (14.3)
Age (mean ± SD)	71 ± 12	70 ± 15	69 ± 9	74 ± 6
Underlying heart disease (N, %)				
CAD	24 (68.6)	4 (28.6)	6 (25)	6 (42.9)
AVD	2 (5.7)	2 (14.3)	3 (12.5)	3 (21.4)
MVD	2 (5.7)	3 (21.4)	11 (45.8)	3 (21.4)
CAD + AVD	6 (17.1)	3 (21.4)	3 (12.5)	2 (14.3)
CAD + MVD	1 (2.9)	2 (14.3)	1 (4.2)	0 (0.0)
Duration of AF (mean ± SD (months))	-	89 ± 95	61 ± 53	154 ± 90
LA dilatation (>45 mm, %)	7 (20)	3 (21.4)	13 (54.2)	11 (78.6)
LVF (N, %)				
Normal	29 (82.9)	11 (78.6)	12 (50)	9 (64.3)
Mild impairment	6 (17.1)	2 (14.3)	6 (25)	4 (28.6)
Moderate impairment	0 (0.0)	1 (7.1)	5 (20.8)	1 (7.1)
Severe impairment	0 (0.0)	0 (0.0)	1 (4.2)	0 (0.0)
Medication (N, %)				
ACE	24 (68.6)	8 (57.1)	16 (66.7)	12 (85.7)
Statin	29 (82.9)	8 (57.1)	7 (29.2)	10 (71.4)
Type I AAD	0 (0.0)	2 (14.3)	0 (0.0)	0 (0.0)
Type II AAD	24 (68.6)	7 (50)	17 (70.8)	12 (85.7)
Type III AAD	0 (0.0)	5 (35.7)	3 (12.5)	1 (7.1)
Type IV AAD	0 (0.0)	0 (0.0)	1 (4.2)	1 (7.1)
Digoxin	0 (0.0)	1 (7.1)	8 (33.3)	4 (28.6)

SR: Sinus rhythm, PAF: Paroxysmal AF, PeAF: Persistent AF, LSPeAF: Longstanding persistent AF, RAA: Right atrial appendage, LAA: Left atrial appendage, CAD: Coronary artery disease, AVD: Aortic valve disease, MVD: Mitral valve disease, LA: Left atrium, LVF: Left ventricular function, ACE: Angiotensin-converting enzyme inhibitor, AAD: Anti-arrhythmic drug.

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
