# Peer review of "Mitochondrial Dysfunction Underlies Cardiomyocyte Remodeling in Experimental and Clinical Atrial Fibrillation"

_cells, 2019, doi:10.3390/cells8101202_

Round 1

Reviewer 1 Report

Although this reviewer warmly welcomes this manuscript, some concerns should be addressed:

The rationale for the study is unclear as the introduction is a bit confusing, comprising several pieces of apparently unlinked information. A more integrated appraisal of the relevant literature would be appropriate to provide the context for the study.

The functional role of mitochondria in arrhythmogenesis (Gambardella et al. Adv Exp Med Biol. 2017;982:191-202. doi: 10.1007/978-3-319-55330-6_10) should be better discussed.

The Authors should provide a measurement of mitochondrial ROS (e.g. MitoSOX: Xie et al. Sci Rep. 2015 Jul 14;5:11427).

Some sections within the manuscript read a bit dense with disjointed information and would benefit from re-organization of paragraphs.

The strengths (and limitations) of the study should be deeply addressed, taking into account sources of potential bias or imprecision: Discuss both direction and magnitude of any potential bias.

Reviewer 2 Report

This study evaluated the short-term tachy-pacing in HL-1 cell mitochondria and mitochondrial function from Paf, PeAF and long lasting AF patients or in animal model. This study provided some interesting findings. However, this reviewer has some uncertainty on the study design and data interpretation, since this study only included 6-8 hours pacing model. It is not clear whether long-term (more than 1 day) may result in the same findings. In addition, there were huge different expression between different Af patients. Do different pattern of AF in humans lead to different mitochondria or the mitochondrial differences lead to different patterns?

Specific comments

Please show the experimental number in all presented data. It is difficult to check the data without experimental number. Please show the image for calcium transient or respiratory and oxidative stress. Is there any action potential morphology changed or cytoplasmic calcium homeostasis the studied cells? Dese the ATP deficient shorten the AP duration via action of IKATP? Why did the maximum respiration increase at 6 hours and returns to normal at 8 hours? The pattern was completely different from those in other mitochondrial functions. The western blot in Figure 1I, IK were poor on density resolution. It seems to be uncountable. Why are the western blot in human samples fragmented? Is there mechanistic insight on the differences between RA and LA especially in different group patients? What is role of HSP in the mitochondria? It sounds likely very speculative.

Reviewer 3 Report

In this manuscript, Wiersma et al present findings that propose mitochondrial dysfunction as modulator of AF progression using paced HL-1 cells and Drosophila, as well as AF patient biopsies.  Using genetic and chemical tools, the authors conclude that tachypacing induces mitochondrial dysfunction that may be mediated through MCU. Overall, the data are mostly well-presented, however some conclusions are not fully supported and should be addressed prior to acceptance for publication.

Imaging of mitochondrial morphology: Control panel in Figure 4B is the exact same image as Figure S6.  A more representative image of non-paced morphology is should be used, as the current control image appears blurry and out of focus. Along these lines, it is difficult to distinguish between the intermediate and fragmented phenotypes.  Indications would help the reader. Conclusions about the role of MCU: The conclusion that tachypacing-induced changes are directly due to MCU is unsupported. The data support that it is one change that may be altered following tachypacing.  Can early changes in MCU expression observed following tachypacing?  There doesn’t appear to be any changes in MCU expression in 6hTP or 8hTP (Fig. 5a).  If this is an acute response, as has been shown in mice, an earlier time point should be tested.  Do expression changes precede the onset of mitochondrial dysfunction or is it simply a consequence of it? How has the MCU OE line been validated? Localization of the OE construct must be examined to show that construct is targeted to the mitochondria, and could be performed using the V5 tag.  What is the relative MCU expression level in OE compared to the control and siRNA conditions? How were 60% and 80% KD achieved? Not described in the methods clearly. The authors use a tachypaced Drosophila model, but it is not shown in great detail. Do RU360 treatments in Drosophila similarly increase HSP60 and HSP10 mRNA levels?  What is the mitochondrial morphology in these animals?

Minor Comments:

Correct nomenclature rules need to be followed (i.e. italicized names for mRNA)

Reviewer 4 Report

This submission "Mitochondrial dysfunction underlies cardioyocyte remodelin in experimental and clinical atrial fibrillation" uses a cell based model for cardiac dysfunction, drosophila melanogaster and human patient samples to explore the role of mitochondrial stress on cardiac dysfunction.  While neither model system appears to be a perfect model for atrial fibrillation, interesting findings in these model system may be applicable further human studies.

Major concerns to be address

Fluo4 data is mentioned in the materials and methods but I did not see this data in the text.  Since the use of Rhod2 as a mitochondrial specific Calcium indicator is somewhat controversial.  The authors should show the Fluo4 data.  They should also include references to justify using Rhod2 as a mitochondrial specific Calcium reporter.

Missing control (SR) data for LAA in figure 2 makes these finding impossible to interpret.

Inclusion data for patients in the peAF v. LSPeAF group should be spelled out. What was the time cut off for including patients in one group vs. the other.  In Table 1 it appears that there could be overlap between these two groups based on the duration data.  also could the authors please explain the duration data for paroxysmal AF.  Since the duration of PAF appears to be longer than for the PeAF group does this indicate that these patients have had multiple procedures to reverse AF?

Also from Table 1  There are very large differences in the medication regimes of patients in the various groups.  Could the other discuss whether these differences might impact their finding?  Might the mechanism of action for any of these drugs impact mitochondrial function?

In figure 3, both TEMPO  appears to exacerbate the phenotype of tachypacing.  Could the others please comment on this in the text and discuss the mechanism.

Also in figure 3 low does of Ru360 exacerabate the CaT phenotype of tachypacing whereas higher doses improve this phenotype. Could the authors comment on this finding.

Minor points:

There were some minor spelling/grammar errors throughout the text (e.g. it is  Lipofectamine  not lipofectamin

In figures 4B and 6J images of mitochondrial morphology are not publication quality.  Also, the controls are identical.  The authors might consider merging 4B and 6J into a single figure or use different control images.

Throughuot, figures should have legends within the figure itself

Round 2

Reviewer 1 Report

-

Author Response

We thank reviewer 1 for his/her approval of our manuscript.

Reviewer 2 Report

The authors have addressed my concerns. I did not have more questions. 

Author Response

We thank reviewer 4 for his/her approval of our manuscript.

Reviewer 3 Report

While the authors have made some adjustments to strengthen the manuscript, the following points raised are not adequately addressed.

Image of mitochondrial morphology (comment 1): FigS6 - the normal paced HL-1 CM image still looks blurry and it remains the same as the previous version.  It also is still unclear what the authors count as tubular and fragmented mitochondria.  A highlight or schematic to indicate what the two morphologies is strongly recommended.

MCU OE line validation (comment 3).  Authors response indicates a WB should be in FIgure S9 - it is not there.  Furthermore, a WB doesn't answer the point regarding localization to the mitochondria.  Again, immunohistochemistry is suggested.  Without this, it is hard to say whether the overexpression line is functional or not, and thus the conclusion that MCU OE has no effect is not supported.

Author Response

Image of mitochondrial morphology (comment 1): FigS6 - the normal paced HL-1 CM image still looks blurry and it remains the same as the previous version. It also is still unclear what the authors count as tubular and fragmented mitochondria. A highlight or schematic to indicate what the two morphologies is strongly recommended.

Thank you for your feedback. We have changed the image of the normal-paced HL-1 cardiomyocyte, as suggested by this reviewer.To clarify what we count as tubular, intermediate and fragmented mitochondria, we adapted the methods section (page 4):

‘Ten random fields containing at least 10 cardiomyocytes were selected and the mitochondrial morphology per single cardiomyocyte was scored as tubular, intermediate or fragmented by an investigator blinded for the conditions. The network was scored as depicted and described in [21]. Briefly, the network was scored as tubular when it appeared as long, intertwining tubules; as intermediate when the tubules were at least 30% shorter and also showed dots (single mitochondria) in between; and as fragmented when >70% of the network were dots instead of tubules. The amount of tubular, intermediate or fragmented cardiomyocytes were expressed as percentage of the total cardiomyocytes to show the distribution of tubular, intermediate or fragmented mitochondrial morphology between conditions.’

MCU OE line validation (comment 3). Authors response indicates a WB should be in Figure S9 - it is not there. Furthermore, a WB doesn't answer the point regarding localization to the mitochondria. Again, immunohistochemistry is suggested. Without this, it is hard to say whether the overexpression line is functional or not, and thus the conclusion that MCU OE has no effect is not supported.

Thank you for your comment. In Figure S9A, we show the quantified data of the Western blot with HL-1 cardiomyocytes with and without MCU OE stained for MCU and corrected for GAPDH values. This graph shows that MCU OE has increased MCU expression. Although changes in cardiomyocyte contractile function were observed between MCU OE compared to control, indicating a functional effect of MCU OE, we did, however, not determine whether the MCU OE is localized in the mitochondria. Since this can be considered a limitation of this study, we added to the Strengths and limitations paragraph in the discussion section (page 19):

‘The MCU OE plasmid showed an increased MCU protein expression, however, we did not determine the localization to the mitochondria. Nevertheless, functionally we show that MCU OE and downregulating of MCU, by siRNA, has a detrimental and beneficial effect, respectively on CaTmito, indicating that MCU modulates mitochondrial calcium.’

Reviewer 4 Report

The authors have satisfied my major concerns regarding this study and have altered the text  where appropriate to acknowledge unavoidable flaws in the study.

Author Response

(The authors gave the same response as above.)
